# Learning Robust Options
# by Conditional Value at Risk Optimization

**Takuya Hiraoka** [1,2,3], **Takahisa Imagawa** [2], **Tatsuya Mori** [1,2,3], **Takashi Onishi** [1,2],
**Yoshimasa Tsuruoka** [2,4]

[1]NEC Corporation
[2]National Institute of Advanced Industrial Science and Technology
[3]RIKEN Center for Advanced Intelligence Project
[4]The University of Tokyo
`{takuya-h1, tmori, takashi.onishi}@nec.com`
`imagawa.t@aist.go.jp, tsuruoka@logos.t.u-tokyo.ac.jp`

## Abstract

Options are generally learned by using an inaccurate environment model (or simulator), which contains uncertain model parameters. While there are several methods to learn options that are robust against the uncertainty of model parameters, these methods only consider either the worst case or the average (ordinary) case for learning options. This limited consideration of the cases often produces options that do not work well in the unconsidered case. In this paper, we propose a conditional value at risk (CVaR)-based method to learn options that work well in both the average and worst cases. We extend the CVaR-based policy gradient method proposed by Chow and Ghavamzadeh (2014) to deal with robust Markov decision processes and then apply the extended method to learning robust options. We conduct experiments to evaluate our method in multi-joint robot control tasks (HopperIceBlock, Half-Cheetah, and Walker2D). Experimental results show that our method produces options that 1) give better worst-case performance than the options learned only to minimize the average-case loss, and 2) give better average-case performance than the options learned only to minimize the worst-case loss.

## 1 Introduction

In the reinforcement learning context, an *Option* means a temporally extended sequence of actions [30], and is regarded as useful for many purposes, such as speeding up learning, transferring skills across domains, and solving long-term planning problems. Because of its usefulness, the methods for discovering (learning) options have been actively studied (e.g., [1, 17, 18, 20, 28, 29]).

Learning options successfully in practical domains requires a large amount of training data, but collecting data from a real-world environment is often prohibitively expensive [15]. In such cases, environment models (i.e., simulators) are used for learning options instead, but such models usually contain uncertain parameters (i.e., a reality gap) since they are constructed based on insufficient knowledge of a real environment. Options learned with an inaccurate model can lead to a significant degradation of the performance of the agent when deployed in the real environment [19]. This problem is a severe obstacle to applying the options framework to practical tasks in the real world, and has driven the need for methods that can learn robust options against inaccuracy of models.

Some previous work has addressed the problem of learning options that are robust against inaccuracy of models. Mankowitz et al. [19] proposed a method to learn options that minimize the expected loss (or maximize an expected return) on the environment model with the worst-case model parameter

values, which are selected so that the expected loss is maximized. Frans et al. [10] proposed a method to learn options that minimize the expected loss on the environment model with the average-case model parameter values.

However, these methods only consider either the worst-case model parameter values or the average-case model parameter values when learning options, and this causes learned options to work poorly in an environment with the unconsidered model parameter values. Mankowitz's [19] method produces options that are overly conservative in the environment with an average-case model parameter value [9]. In contrast, Frans's method [10] produces options that can cause a catastrophic failure in an environment with the worst-case model parameter value.

Furthermore, Mankowitz's [19] method does not use a distribution of the model parameters for learning options. Generally, the distribution can be built on the basis of prior knowledge. If the distribution can be used, as in Derman et al. [9] and Rajeswaran et al. [24], one can use it to adjust the importance of the model parameter values in learning options. However, Mankowitz's method does not consider the distribution and always produces the policy to minimize the loss in the worst case model parameter value. Therefore, even if the worst case is extremely unlikely to happen, the learned policy is adapted to the case and results in being overly conservative.

To mitigate the aforementioned problems, we propose a conditional value at risk (CVaR)-based method to learn robust options optimized for the expected loss in both the average and worst cases. In our method, an option is learned so that it 1) makes the expected loss lower than a given threshold in the worst case and also 2) decreases the expected loss in the average case as much as possible. Since our method considers both the worst and average cases when learning options, it can mitigate the aforementioned problems with the previous methods for option learning. In addition, unlike Mankowitz's [19] method, our method evaluates the expected loss in the worst case on the basis of CVaR (see Section 3). This enables us to reflect the model parameter distribution in learning options, and thus prevents the learned options from overfitting an environment with an extremely rare worst-case parameter value.

Our main contributions are as follows: **(1)** We introduce CVaR optimization to learning robust options and illustrate its effectiveness (see Section 3, Section 4, and Appendices). Although some previous works have proposed CVaR-based reinforcement learning methods (e.g., [6, 24, 32]), their methods have been applied to learning robust "flat" policies, and option learning has not been discussed. **(2)** We extend Chow and Ghavamzadeh's CVaR-based policy gradient method [6] so that it can deal with robust Markov decision processes (robust MDPs) and option learning. Their method is for ordinary MDPs and does not take the uncertainty of model parameters into account. We extend their method to robust MDPs to deal with the uncertain model parameters (see Section 3). Further, we apply the extended method to learn robust options. For this application, we derive the option policy gradient theorem [1] to minimise soft robust loss [9] (see Appendices). **(3)** We evaluate our robust option learning methods in several nontrivial continuous control tasks (see Section 4 and Appendices). Robust option learning methods have not been thoroughly evaluated on such tasks. Mankowitz et al. [19] evaluated their methods only on simple control tasks (Acrobot and CartPole with discrete action spaces). Our research is a first attempt to evaluate robust option learning methods on more complex tasks (multi-joint robot control tasks) where state–action spaces are high dimensional and continuous.

This paper is organized as follows. First we define our problem and introduce preliminaries (Section 2), and then propose our method by extending Chow and Ghavamzadeh's method (Section 3). We evaluate our method on multi-joint robot control tasks (Section 4). We also discuss related work (Section 5). Finally, we conclude our research.

## 2 Problem Description

We consider a robust MDP [34], which is a tuple $\langle S, A, \mathbb{C}, \gamma, P, T_p, \mathbb{P}_0 \rangle$ where $S$, $A$, $\mathbb{C}$, $\gamma$, and $P$ are states, actions, a cost function, a discount factor, and an ambiguity set, respectively; $T_p$ and $\mathbb{P}_0$ are a state transition function parameterized by $p \in P$, and an initial state distribution, respectively. The transition probability is used in the form of $s_{t+1} \sim T_p(s_t, a_t)$, where $s_{t+1}$ is a random variable. Similarly, the initial state distribution is used in the form of $s_0 \sim \mathbb{P}_0$. The cost function is used in the form of $c_{t+1} \sim \mathbb{C}(s_t, a_t)$, where $c_{t+1}$ is a random variable. Here, the sum of the costs discounted by $\gamma$ is called a loss $C = \sum_{t=0}^{T} \gamma^t c_t$.

As with the standard robust reinforcement learning settings [9, 10, 24], $p$ is generated by a model parameter distribution $\mathbb{P}(p)$ that captures our subjective belief about the parameter values of a real environment. By using the distribution, we define a *soft robust loss* [9]:

$$\mathbb{E}_{C,p}[C] = \sum_p \mathbb{P}(p)\mathbb{E}_C[C \mid p],\tag{1}$$

where $\mathbb{E}_C[C \mid p]$ is the expected loss on a parameterized MDP $\langle S, A, \mathbb{C}, \gamma, T_p, \mathbb{P}_0 \rangle$, in which the transition probability is parameterized by $p$.

As with Derman et al. [9] and Xu and Mannor [35], we make the rectangularity assumption on $P$ and $\mathbb{P}(p)$. That is, $P$ is assumed to be structured as a Cartesian product $\bigotimes_{s \in S} P_s$, and $\mathbb{P}$ is also assumed to be a Cartesian product $\bigotimes_{s \in S} \mathbb{P}_s(p_s \in P_s)$. These assumptions enable us to define a model parameter distribution independently for each state.

We also assume that the learning agent interacts with an environment on the basis of its policy, which takes the form of the call-and-return option [29]. Option $\omega \in \Omega$ is a temporally extended action and represented as a tuple $\langle I_\omega, \pi_\omega, \beta_\omega \rangle$, in which $I_\omega \subseteq S$ is an initiation set, $\pi_\omega$ is an intra-option policy, and $\beta_\omega : S \to [0, 1]$ is a termination function. In the call-and-return option, an agent picks option $\omega$ in accordance with its policy over options $\pi_\Omega$, and follows the intra-option policy $\pi_\omega$ until termination (as dictated by $\beta_\omega$), at which point this procedure is repeated. Here $\pi_\Omega$, $\pi_\omega$, and $\beta_\omega$ are parameterized by $\theta_{\pi_\Omega}$, $\theta_{\pi_\omega}$, and $\theta_{\beta_\omega}$, respectively.

Our objective is to optimize parameters $\theta = (\theta_{\pi_\Omega}, \theta_{\pi_\omega}, \theta_{\beta_\omega})$ [1] so that the learned options work well in both the average and worst cases. An optimization criterion will be described in the next section.

# 3   Option Critic with the CVaR Constraint

To achieve our objective, we need a method to produce the option policies with parameters $\theta$ that work well in both the average and worst cases. Chow and Ghavamzadeh [6] proposed a CVaR-based policy gradient method to find such parameters in ordinary MDPs. In this section, we extend their policy gradient methods to robust MDPs, and then apply the extended methods to option learning.

First, we define the optimization objective:

$$\min_\theta \mathbb{E}_{C,p}[C] \qquad s.t. \qquad \mathbb{E}_{C,p}[C \mid C \geq \text{VaR}_\epsilon(C)] \leq \zeta,\tag{2}$$

$$\text{VaR}_\epsilon(C) \quad = \quad \max\{z \in \mathbb{R} \mid \mathbb{P}(C \geq z) \geq \epsilon\},\tag{3}$$

where $\zeta$ is the loss tolerance and $\text{VaR}_\epsilon$ is the upper $\epsilon$ percentile (also known as an upper-tail value at risk). The optimization term represents the soft robust loss (Eq. 1). The expectation term in the constraint is CVaR, which is the expected value exceeding the value at risk (i.e., CVaR evaluates the expected loss in the worst case). The constraint requires that CVaR must be equal to or less than $\zeta$. Eq. 2 is an extension of the optimization objective of Chow and Ghavamzadeh [6] to soft robust loss (Eq. 1), and thus it uses the model parameter distribution for adjusting the importance of the expected loss on each parameterized MDP. In the following part of this section, we derive an algorithm for robust option policy learning to minimize Eq. 2 similarly to Chow and Ghavamzadeh [6].

By Theorem 16 in [25], Eq. 2 can be shown to be equivalent to the following objective:

$$\min_{\theta, v \in \mathbb{R}} \mathbb{E}_{C,p}[C] \quad s.t. \quad v + \frac{1}{\epsilon}\mathbb{E}_{C,p}[\max(0, C - v)] \leq \zeta.\tag{4}$$

To solve Eq. 4, we use the Lagrangian relaxation method [3] to convert it into the following unconstrained problem:

$$\max_{\lambda \geq 0} \min_{\theta, v} L(\theta, v, \lambda) \quad s.t. \quad L(\theta, v, \lambda) = \mathbb{E}_{C,p}[C] + \lambda\left(v + \tfrac{1}{\epsilon}\mathbb{E}_{C,p}[\max(0, C - v)] - \zeta\right),\tag{5}$$

where $\lambda$ is the Lagrange multiplier. The goal here is to find a saddle point of $L(\theta, v, \lambda)$, i.e., a point $(\theta^*, v^*, \lambda^*)$ that satisfies $L(\theta, v, \lambda^*) \geq L(\theta^*, v^*, \lambda^*) \geq L(\theta^*, v^*, \lambda), \forall \theta, v, \forall \lambda \leq 0$. This is achieved by descending in $\theta$ and $v$ and ascending in $\lambda$ using the gradients of $L(\theta, v, \lambda)$:

$$\nabla_\theta L(\theta, v, \lambda) = \nabla_\theta \mathbb{E}_{C,p}[C] + \frac{\lambda}{\epsilon} \nabla_\theta \mathbb{E}_{C,p}[\max(0, C - v)], \tag{6}$$

$$\frac{\partial L(\theta, v, \lambda)}{\partial v} = \lambda \left(1 + \frac{1}{\epsilon} \frac{\partial \mathbb{E}_{C,p}[\max(0, C - v)]}{\partial v}\right), \tag{7}$$

$$\frac{\partial L(\theta, v, \lambda)}{\partial \lambda} = v + \frac{1}{\epsilon} \mathbb{E}_{C,p}[\max(0, C - v)] - \zeta. \tag{8}$$

### 3.1 Gradient with respect to the option policy parameters $\theta_{\pi_\Omega}$, $\theta_{\pi_\omega}$, and $\theta_{\beta_\omega}$

In this section, based on Eq. 6, we propose policy gradients with respect to $\theta_{\pi_\Omega}$, $\theta_{\pi_\omega}$, and $\theta_{\beta_\omega}$ (Eq. 12, Eq. 13, and Eq. 14), which are useful for developing an option-critic style algorithm. First, we show that Eq. 6 can be simplified as the gradient of soft robust loss in an augmented MDP, and then derive the policy gradients of it.

By applying Corollary 1 in Appendices and Eq. 1, Eq. 6 can be rewritten as

$$\nabla_\theta L(\theta, v, \lambda) = \sum_p \mathbb{P}(p) \nabla_\theta \left( \mathbb{E}_C[C \mid p] + \frac{\lambda}{\epsilon} \mathbb{E}_C[\max(0, C - v) \mid p] \right). \tag{9}$$

In Eq. 9, the expected soft robust loss term and constraint term are contained inside the gradient operator. If we proceed with the derivation of the gradient while treating these terms as they are, the derivation becomes complicated. Therefore, as in Chow and Ghavamzadeh [6], we simplify these terms to a single term by extending the optimization problem. For this, we augment the parameterized MDP by adding an extra reward factor of which the discounted expectation matches the constraint term. We also extend the state to consider an additional state factor $x \in \mathbb{R}$ for calculating the extra reward factor. Formally, given the original parameterized MDP $\langle S, A, \mathbb{C}, \gamma, T_p, \mathbb{P}_0 \rangle$, we define an augmented parameterized MDP $\langle S', A, \mathbb{C}', \gamma, T'_p, \mathbb{P}'_0 \rangle$ where $S' = S \times \mathbb{R}$, $\mathbb{P}'_0 = \mathbb{P}_0 \cdot \mathbf{1}(x_0 = v)$ [2], and

$$\mathbb{C}'(s', a) = \begin{cases} \mathbb{C}(s, a) + \lambda \max(0, -x)/\epsilon & \text{(if } s \text{ is a terminal state)} \\ \mathbb{C}(s, a) & \text{(otherwise)} \end{cases},$$

$$T'_p(s', a) = \begin{cases} T_p(s, a) & \text{(if } x' = (x - \mathbb{C}(s, a))/\gamma) \\ 0 & \text{(otherwise)} \end{cases}.$$

Here the augmented state transition and cost functions are used in the form of $s'_{t+1} \sim T'_p(s'_t, a_t)$ and $c'_{t+1} \sim \mathbb{C}'(s'_t, a_t)$, respectively. On this augmented parameterized MDP, the loss $C'$ can be written as [3]

$$C' = \sum_{t=0}^T \gamma^t c'_t = \sum_{t=0}^T \gamma^t c_t + \frac{\lambda \max\left(0, \sum_{t=0}^T \gamma^t c_t - v\right)}{\epsilon}. \tag{10}$$

From Eq. 10, it is clear that, by extending the optimization problem from the parameterized MDP into the augmented parameterized MDP, Eq. 9 can be rewritten as

$$\nabla_\theta L(\theta, v, \lambda) = \sum_p \mathbb{P}(p) \nabla_\theta \mathbb{E}_{C'}[C' \mid p]. \tag{11}$$

By Theorems 1, 2, and 3 in Appendices[4], Eq. 11 with respect to $\theta_{\pi_\Omega}$, $\theta_{\pi_\omega}$, and $\theta_{\beta_\omega}$ can be written as

$$\frac{\partial L(\theta, v, \lambda)}{\partial \theta_{\pi_\Omega}} \ = \ \mathbb{E}_{s'}\left[\sum_\omega \frac{\partial \pi_\Omega(\omega|s')}{\partial \theta_{\pi_\Omega}}\overline{Q}_\Omega(s', \omega)\ \middle|\ \overline{p}\right], \tag{12}$$

$$\frac{\partial L(\theta, v, \lambda)}{\partial \theta_{\pi_\omega}} \ = \ \mathbb{E}_{s',\omega}\left[\sum_a \frac{\partial \pi_\omega(a|s')}{\partial \theta_{\pi_\omega}}\overline{Q}_\omega(s', \omega, a)\ \middle|\ \overline{p}\right], \tag{13}$$

$$\frac{\partial L(\theta, v, \lambda)}{\partial \theta_{\beta_\omega}} \ = \ \mathbb{E}_{s',\omega}\left[\frac{\partial \beta_\omega(s')}{\partial \theta_{\beta_\omega}}\left(\overline{V}_\Omega(s') - \overline{Q}_\Omega(s', \omega)\right)\ \middle|\ \overline{p}\right]. \tag{14}$$

Here $\overline{Q}_\Omega$, $\overline{Q}_\omega$, and $\overline{V}_\Omega$ are the average value (i.e., the soft robust loss) of executing option $\omega$, the value of executing an action in the context of a state-option pair, and state value. In addition, $\mathbb{E}_{s'}\left[\bullet \mid \overline{p}\right]$ and $\mathbb{E}_{s',\omega}\left[\bullet \mid \overline{p}\right]$ are the expectations of argument $\bullet$, with respect to the probability of trajectories generated from the average augmented MDP $\langle S', A, \mathbb{C}', \gamma, \mathbb{E}_p\left[T'_p\right], \mathbb{P}'_0\rangle$, which follows the average transition function $\mathbb{E}_p\left[T'_p\right] = \sum_p \mathbb{P}(p)T'_p$, respectively.

## 3.2 Gradient with respect to $\lambda$ and $v$

As in the last section, we extend the parameter optimization problem into an augmented MDP. We define an augmented parameterized MDP $\langle S', A, \mathbb{C}'', \gamma, T'_p, \mathbb{P}'_0\rangle$ where

$$\mathbb{C}''(s', a) \ = \ \begin{cases} \max(0, -x) & \text{(if } s \text{ is a terminal state)} \\ 0 & \text{(otherwise)} \end{cases}, \tag{15}$$

and other functions and variables are the same as those in the augmented parameterized MDP in the last section. Here the augmented cost function is used in the form of $c''_{t+1} \sim \mathbb{C}''(s'_t, a_t)$. On this augmented parameterized MDP, the expected loss $C''$ can be written as

$$C'' = \sum_{t=0}^T \gamma^t c''_t = \max\left(0, \sum_{t=0}^T \gamma^t c_t - v\right). \tag{16}$$

From Eq. 16, it is clear that, by extending the optimization problem from the parameterized MDP to the augmented parameterized MDP, $\mathbb{E}_{C,p}\left[\max\left(0, C - v\right)\right]$ in Eq. 7 and Eq. 8 can be replaced by $\mathbb{E}_{C'',p}\left[C''\right]$. Further, by Corollary 2 in Appendices, this term can be rewritten as $\mathbb{E}_{C'',p}\left[C''\right] = \mathbb{E}_{C''}\left[C'' \mid \overline{p}\right]$.

Therefore, Eq. 7 and Eq. 8 can be rewritten as

$$\frac{\partial L(\theta, v, \lambda)}{\partial v} \ = \ \lambda\left(1 + \frac{1}{\epsilon}\frac{\partial \mathbb{E}_{C''}\left[C'' \mid \overline{p}\right]}{\partial v}\right) = \lambda\left(1 - \frac{1}{\epsilon}\mathbb{E}_{C''}\left[\mathbf{1}\left(C'' \geq 0\right) \mid \overline{p}\right]\right), \tag{17}$$

$$\frac{\partial L(\theta, v, \lambda)}{\partial \lambda} \ = \ v + \frac{1}{\epsilon}\mathbb{E}_{C''}\left[C'' \mid \overline{p}\right] - \zeta. \tag{18}$$

## 3.3 Algorithmic representation of Option Critic with CVaR Constraints (OC3)

In the last two sections, we derived the gradient of $L(\theta, v, \lambda)$ with respect to the option policy parameters, $v$, and $\lambda$. In this section, on the basis of the derivation result, we propose an algorithm to optimize the option policy parameters.

Algorithm 1 shows a pseudocode for learning options with the CVaR constraint. In lines 2–5, we sample trajectories to estimate state-action values and find parameters. In this sampling phase, analytically finding the average transition function may be difficult, if the transition function is given as a black-box simulator. In such cases, sampling approaches can be used to find such average transition functions. One possible approach is to sample $p$ from $\mathbb{P}(p)$, and approximate $\mathbb{E}_p\left[T'_p\right]$ by a sampled transition function. Another approach is to, for each trajectory sampling, sample $p$ from

**Algorithm 1** Option Critic with CVaR Constraint (OC3)

**Input:** $\theta_{\pi_\omega,0}, \theta_{\beta_\omega,0}, \theta_{\pi_\Omega,0}, v_0, \lambda_0, N_{iter}, N_{epi}, \zeta, \Omega$
1: **for** iteration $i = 0, 1, ..., N_{iter}$ **do**
2:     **for** $k = 0, 1, ..., N_{epi}$ **do**
3:         Sample a trajectory $\tau_k = \left\{ s'_t, \omega_t, a_t, (c'_t, c''_t), s'_{t+1} \right\}_{t=0}^{T-1}$ from the average augment MDP
        $\left\langle S', A, (\mathbb{C}', \mathbb{C}''), \gamma, \mathbb{E}_p\left[T'_p\right], \mathbb{P}'_0 \right\rangle$ by option policies $(\pi_\omega, \beta_\omega, \pi_\Omega)$ with $\theta_{\pi_\omega,i}, \theta_{\beta_\omega,i}$, and $\theta_{\pi_\Omega,i}$.
4:     **end for**
5:     Update $\overline{Q}_\Omega$, and $\overline{V}_\Omega$ with $\left\{\tau_0, ..., \tau_{N_{epi}}\right\}$.
6:     **for** $\omega \in \Omega$ **do**
7:         Update $\overline{Q}_\omega$ with $\left\{\tau_0, ..., \tau_{N_{epi}}\right\}$.
8:         $\theta_{\pi_\omega,i+1} \leftarrow \theta_{\pi_\omega,i} - \alpha \frac{\partial L(\theta_i, v_i, \lambda_i)}{\partial \theta_{\pi_\omega,i}}$
9:         $\theta_{\beta_\omega,i+1} \leftarrow \theta_{\beta_\omega,i} - \alpha \frac{\partial L(\theta_i, v_i, \lambda_i)}{\partial \theta_{\beta_\omega,i}}$
10:    **end for**
11:    $\theta_{\pi_\Omega,i+1} \leftarrow \theta_{\pi_\Omega,i} - \alpha \frac{\partial L(\theta_i, v_i, \lambda_i)}{\partial \theta_{\pi_\Omega,i}}$
12:    $v_{i+1} \leftarrow v_i - \alpha \frac{\partial L(\theta_i, v_i, \lambda_i)}{\partial v_i}$
13:    $\lambda_{i+1} \leftarrow \lambda_i + \alpha \frac{\partial L(\theta_i, v_i, \lambda_i)}{\partial \lambda_i}$
14: **end for**

$\mathbb{P}(p)$ and then generate a trajectory from the augment parameterized MDP $\left\langle S', A', (\mathbb{C}', \mathbb{C}''), \gamma, T'_p, \mathbb{P}'_0 \right\rangle$. In lines 6–11, we update the value functions, and then update the option policy parameters on the basis of on Eq. 12, Eq. 13, and Eq. 14. The implementation of this update part differs depending on the base reinforcement learning method. In this paper, we adapt the proximal policy option critic (PPOC) [16] to this part [5]. In lines 12–14, we update $v$ and $\lambda$. In this part, the gradient with respect to $v$ and $\lambda$ are calculated by Eq. 17 and Eq. 18.

Our algorithm is different from the Actor-Critic Algorithm for CVaR Optimization [6] mainly in the following points:
**Policy structure :** In the algorithm in [6], a flat policy is optimized, while, in our algorithm, the option policies, which are more general than the flat policy [6], are optimized.
**The type of augmented MDP:** In the algorithm in [6], the augmented MDP takes the form of $\left\langle S', A, \mathbb{C}', \gamma, T', \mathbb{P}'_0 \right\rangle$, while, in our algorithm, the augmented MDP takes the form of $\left\langle S', A, \mathbb{C}', \gamma, \mathbb{E}_p\left[T'_p\right], \mathbb{P}'_0 \right\rangle$. Here $T'$ is the transition function without model parameters. This difference ($T'$ and $\mathbb{E}_p\left[T'_p\right]$) comes from the difference of optimization objectives. In [6], the optimization objective is the expected loss without a model parameter uncertainty ($\mathbb{E}_{C'}[C']$), while in our optimization objective (Eq. 2), the model parameter uncertainty is considered by taking the expectation of the model parameter distribution ($\mathbb{E}_{C',p}[C'] = \sum_p \mathbb{P}(p)\mathbb{E}_{C'}[C' \mid p]$).

## 4 Experiments

In this section, we conduct an experiment to evaluate our method (OC3) on multi-joint robot control tasks with model parameter uncertainty [7]. Through this experiment, we elucidate answers for the following questions:
**Q1:** Can our method (OC3) successfully produce options that satisfy constraints in Eq. 2 in non-trivial cases? Note that since $\zeta$ in Eq. 2 is a hyper-parameter, we can set it to an arbitrary value. If we use a very high value for $\zeta$, the learned options easily satisfy the constraints. Our interest lies in the non-trivial cases, in which the $\zeta$ is set to a reasonably low value.
**Q2:** Can our method successfully produce options that work better in the worst case than the options that are learned only to minimize the average-case loss (i.e., soft robust loss Eq. 1)?

**Q3:** Can our method successfully produce options that work better in the average case than the options that are learned only to minimize worst-case losses? Here the worst case losses are the losses in the worst case. The expectation term of the constraint in Eq. 2 is an instance of such losses. Another instance is the loss in an environment following the worst-case model parameter:

$$\mathbb{E}_C \left[ C \mid \tilde{p} \right] \quad s.t. \quad \tilde{p} = \arg \max_{p \in P} \mathbb{E}_C \left[ C \mid \tilde{p} \right]. \tag{19}$$

In this experiment, we compare our proposed method with three baselines. All methods are implemented by extending PPOC [16].
**SoftRobust:** In this method, the option policies are learned to minimize an expected soft robust loss (Eq. 1) the same as in Frans et al. [10]. We implemented this method, by adapting the meta learning shared hierarchies (MLSH) algorithm [10] [8] to PPOC.
**WorstCase:** In this method, the option policies are learned to minimize the expected loss on the worst case (i.e., Eq. 19) as in Mankowitz et al. [19]. This method is implemented by modifying the temporal difference error evaluation in the generalized advantage estimation [26] part of PPOC, as $-V(s_t) + c_{t+1} + \gamma \max_{p \in P} \left[ V \left( s_{t+1} \sim T_p (s_t, a_t) \right) \right]$. If the model parameters are continuous, $P$ becomes an infinite set and thus evaluating this error becomes intractable. To mitigate this intractability for the continuous model parameter, we discretize the parameters by tile-coding and use the discretized parameter set as $P$.
**EOOpt-$\epsilon$:** In this method, the option policies are learned to minimize the expected loss in the worst case. Unlike WorstCase, in this method, the expectation term of the constraint in Eq. 2 is used for the optimization objective. This method is implemented by adapting the EPOpt-$\epsilon$ algorithm [24] to PPOC (see Algorithm 2 in Appendices). In this experiment, we set the value of $\epsilon$ to 0.1.
**OC3:** In this method, the option policies are learned to minimize the expected loss on the average augmented MDP while satisfying the CVaR constraints (i.e., Eq. 4). This method is implemented by applying Algorithm 1 to PPOC. In this experiment, $\epsilon$ is set to 0.1, and $\zeta$ is determined so that the produced options achieve a CVaR score that is equal to or better than the score achieved by the best options produced by the other baselines (we will explain this in a later paragraph).
For all of the aforementioned methods, we set the hyper-parameters (e.g., policy and value network architecture and learning rate) for PPOC to the same values as in the original paper [16]. The parameters of the policy network and the value network are updated when the total number of trajectory reaches 10240. This parameter update is repeated 977 times for each learning trial.

The experiments are conducted in the robust MDP extension of the following environments:
**Half-Cheetah:** In this environment, options are used to control the half-cheetah, which is a planar biped robot with eight rigid links, including two legs and a torso, along with six actuated joints [33].
**Walker2D:** In this environment, options are used to control the walker, which is a planar biped robot consisting of seven links, corresponding to two legs and a torso, along with six actuated joints [5].
**HopperIceBlock:** In this environment, options are used to control the hopper, which is a robot with four rigid links, corresponding to the torso, upper leg, lower leg, and foot, along with three actuated joints [13, 16]. The hopper has to pass the block either by jumping completely over them or by sliding on their surface. This environment is more suitable for evaluating the option framework than standard Mujoco environments [5] since it contains explicit task compositionality.
To extend these environments to the robust MDP setting (introduced in Section 2), we change the environment so that some model parameters (e.g., the mass of robot's torso and ground friction) are randomly initialized in accordance with model parameter distributions. For the model parameter distribution, we prepare two types of distribution: continuous and discrete. For the continuous distribution, as in Rajeswaran et al. [24], we use a truncated Gaussian distribution, which follows the hyperparameters described in Table 1 in Appendices. For the discrete distribution, we use a Bernoulli distribution, which follows hyperparameters described in Table 2 in Appendices. We define the cost as the negative of the reward retrieved from the environment.

Regarding **Q1**, we find that OC3 can produce feasible options that keep CVaR lower than a given $\zeta$ (i.e., it satisfies the constraint in Eq. 2). The negative CVaR of each method in each environment is shown in Figure 1 (b). To focus on the non-trivial cases, we choose the best CVaR of the baselines (SoftRobust, WorstCase, and EOOpt-$\epsilon$) and set $\zeta$ to it. For example, in the "Walker2D-disc" environment, the CVaR of WorstCase is used for $\zeta$. The "OC3" in Figure 1 (b) is the negative CVaR with the given $\zeta$. We can see that the negative CVaR score of OC3 is higher than the best negative

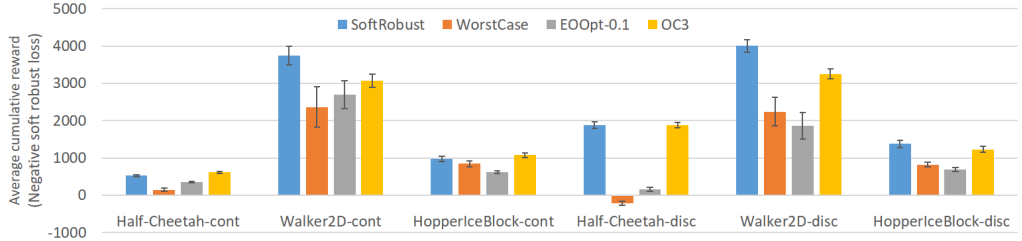

(a) The average-case performance: average cumulative reward (**the negative of the soft robust loss** Eq. 1)

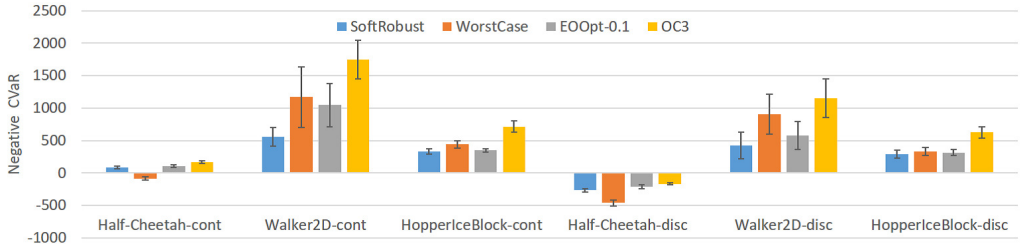

(b) The worst-case performance: negative CVaR with $\epsilon = 0.1$. Here, CVaR is calculated as the average of losses included in the upper 10th percentile, and the negative of it is shown to the figure.

Figure 1: Comparison of methods in the environments. In (a), the vertical axis represents the average cumulative reward (**the negative of the soft robust loss**) of each method. In (b), the vertical axis represents **the negative of CVaR** of each method. In both (a) and (b), the horizontal axis represents environments where the methods are evaluated. The environments with suffix "-cont" are environments with the continuous model parameter distributions, and the environments with "-disc" are environments with the discrete model parameter distributions. The methods with high score values can be regarded as better. In addition, each score is averaged over 36 learning trials with different initial random seeds, and the 95% confidence interval is attached to the score.

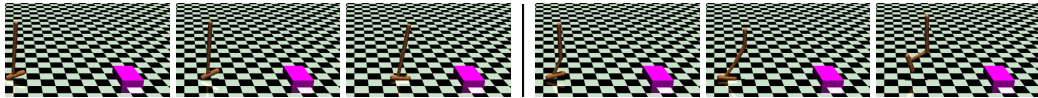

Figure 2: Learned options in HopperIceBlock-disc. Left: the option for walking on the slippery ground. Right: the option for jumping onto the box.

CVaR score the baselines in each environment (i.e., the CVaR score of OC3 is lower than the given $\zeta$). These results clearly demonstrate that our method successfully produces options satisfying the constraint. In addition, the numbers of successful learning trials for OC3 also support our claim (see Table 4 in Appendices).

In addition, regarding **Q2**, Figure 1 (b) also shows that the negative CVaRs of OC3 are higer than those of SoftRobust, in which the option is learned only to minimize the loss in the average case. Therefore, our method can be said to successfully produce options that work better in the worst case than the options learned only for the average case.

Regarding **Q3**, we compare the average-case (soft robust) loss (Eq. 1) of OC3 and those of worst-case methods (WorstCase and EOOpt-0.1). The soft robust loss of each method in each environment is shown in Figure 1 (a). We can see that the scores of OC3 are higher than those of WorstCase and EOOpt-0.1. These results indicate that our method can successfully produce options that work better in the average case than the options learned only for the worst case.

In the analysis of learned options, we find that OC3 and SoftRobust successfully produce options corresponding to decomposed skills required for solving the tasks in most environments. Especially, OC3 produces robust options. For example, in HopperIceBlock-disc, OC3 produces options corresponding to walking on slippery grounds and jumping onto a box (Figures 2). In addition, in Half-Cheetah-disc, OC3 produces an option for running (highlighted in green in Figure 3) and an option for stabilizing the cheetah-bot's body (highlighted in red in Figure 3), which is used mainly in the rare-case model parameter setups. Acquisition of such decomposed robust skills is useful when transferring to a different domain, and they can be also used for post-hoc human analysis and maintenance, which is an important advantage of option-based reinforcement learning over flat-policy reinforcement learning.

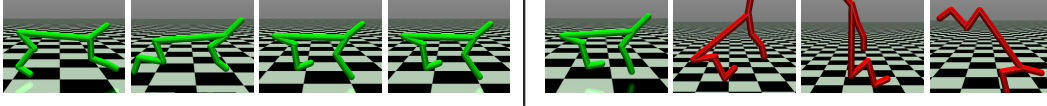

Figure 3: Learned options (shown in green when option 1 is taking the control and in red otherwise) in Half-Cheetah-cont. Left: the case with an ordinary model parameter setup. Right: the case with a rare model parameters setup.

## 5 Related Work

**Robust reinforcement learning (worst case):** One of the main approaches for robust reinforcement learning is to learn policies by minimizing the expected loss in the worst case. "Worst case" is used in two different contexts: 1) the worst case under coherent uncertainty and 2) the worst case under parameter uncertainty. Coherent uncertainty is induced by the stochastic nature of MDPs (i.e., stochastic rewards and transition), whereas parameter uncertainty is induced by inaccurate environment models. For the worst case under coherent uncertainty, many robust reinforcement learning approaches have been proposed (e.g., [7, 8, 12]). Also, many robust reinforcement learning methods have been proposed for the worst case under parameter uncertainty [2, 14, 22, 23, 31, 34]. These works focus on learning robust "flat" policies (i.e., non-hierarchical policies), whereas we focus on learning robust "option" policies (i.e., hierarchical policies).

**Robust reinforcement learning (CVaR):** CVaR has previously been applied to reinforcement learning. Boda and Filar [4] used a dynamic programming approach to optimize CVaR. Morimura et al. [21] introduce CVaR to the exploration part of the SARSA algorithm. Rajeswaran et al. [24] introduced CVaR into model-based Bayesian reinforcement learning. Tamar and Chow [6, 32] presented a policy gradient theorem for CVaR optimizing. These research efforts focus on learning robust flat policies, whereas we focus on learning robust options.

**Learning robust options:** Learning robust options is a relatively recent topic that not much work has addressed. Mankowitz et al. [19] are pioneers of this research topic. They proposed a robust option policy iteration to learn an option that minimizes the expected loss in the worst case under parameter uncertainty. Frans et al. [10] proposed a method to learn options that maximize the expected loss in the average case[9]. Both methods consider loss in either the worst or average case. Mankowitz's method considers only the loss in the worst case, so the produced options are overly adapted to the worst case and does not work well in the average case. On the other hand, Frans's method considers only the loss in the average case, so the produced options work poorly in the unconsidered case. In contrast to these methods, our method considers the losses in both the average and worst cases and thus can mitigate the aforementioned problems.

## 6 Conclusion

In this paper, we proposed a conditional value at risk (CVaR)-based method to learn options so that they 1) keep the expected loss lower than the given threshold (loss tolerance $\zeta$) in the worst case and also 2) decrease the expected loss in the average case as much as possible. To achieve this, we extended Chow and Ghavamzadeh's CVaR policy gradient method [6] to adapt robust Markov decision processes (robust MDPs) and then applied the extended method to learn robust options. For application, we derived a theorem for an option policy gradient for soft robust loss minimization [9]. We conducted experiments to evaluate our method in multi-joint robot control tasks. Experimental results show that our method produces options that 1) work better in the worst case than the options learned only to minimize the loss in the average case and 2) work better in the average case than the options learned only to minimize the loss in the worst case.

Although, in general, the model parameter distribution is not necessarily correct and thus needs to be updated to be more precise by reflecting observations retrieved from the real environment [24], our current method does not consider such model adaptation. One interesting direction for future works is to introduce this model adaptation by extending our method to Bayes-Adaptive Markov decision processes [11].

## Footnotes

[1] For simplicity, we assume that the parameters $\theta_{\pi_\Omega}, \theta_{\pi_\omega}, \theta_{\beta_\omega}$ are scalar variables. All the discussion in this paper can be extended to the case of multi-dimensional variables by making the following changes: 1) define $\theta$ as the concatenation of the multidimensional extensions of the parameters, and 2) replace all partial derivatives with respect to scalar parameters with vector derivatives with respect to the multidimensional parameters.

[2] $\mathbf{1}(\bullet)$ is an indicator function, which returns the value one if the argument $\bullet$ is true and zero otherwise.

[3] Note that $x_T = -\sum_{t=0}^T \gamma^{-T+t} c_t + \gamma^{-T} v$, and thus $\gamma^T \max(0, -x_T) = \max\left(0, \sum_{t=0}^T \gamma^t c_t - v\right)$

[4]To save the space of main content pages, we describe the detail of these theorems and these proofs in Appendices. Note that the soft robust loss (a model parameter uncertainty) is not considered in the policy gradient theorems for the vanilla option critic framework [1], and that the policy gradient theorems of the soft robust loss (Theorems 1, 2, and 3) are newly proposed in out paper.

[5]In PPOC, the Proximal Policy Optimization method is applied to update parameters [27]. We decided to use PPOC since it produced high-performance option policies in continuous control tasks.

[6]By using single option, the options framework can be specified to a flat policy.

[7]Source code to replicate the experiments is available at `https://github.com/TakuyaHiraoka/Learning-Robust-Options-by-Conditional-Value-at-Risk-Optimization`

[8]To make a fair comparison, "warmup period" and reinitialize $\theta_\Omega$ just after "repeat" in MLSH are avoided.

[9]Although their method is presented as a transfer learning method, their optimization objective is essentially equivalent to the "soft robust" objective in the robust reinforcement learning literature [9]. Thus, we regard their method as a robust option learning framework.

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
