[Supplementary Material]

# Appendices

## Algorithmic Representation of EOOpt-$\epsilon$

The outline of EOOpt-$\epsilon$ is summarized in Algorithm 2. Here $C(\tau_k) = \sum_{t=0}^{T} \gamma^t c_t^{(k)}$ is a loss in the trajectory $k$ and $c_t^{(k)}$ is a cost at the $t$-th turn in $k$. PPOCUpdateParams returns the updated parameters on the basis of given trajectories and parameters. This functions is identical to the advantage estimation and parameter update part of Algorithm 1 in Klissarov et al. [16].

---

**Algorithm 2** EOOpt-$\epsilon$

---

**Input:** $\theta_{\pi_\omega,0}, \theta_{\beta_\omega,0}, \theta_{\pi_\Omega,0}, N_{iter}, N_{epi}, \epsilon, \Omega$
1: **for** iteration $i = 0, 1, ..., N_{iter}$ **do**
2:     **for** $k = 0, 1, ..., N_{epi}$ **do**
3:         sample model parameters $p_k \sim \mathbb{P}$
4:         sample a trajectory $\tau_k = \{s_t, \omega_t, a_t, c_t, s_{t+1}\}_{t=0}^{T-1}$ from parameterized MDPs $\langle S, A, \mathbb{C}, \gamma, T_{p_k}, \mathbb{P}_0 \rangle$ using option policies ($\pi_\omega, \beta_\omega$, and $\pi_\Omega$) with $\theta_{\pi_\omega,i}, \theta_{\beta_\omega,i}$, and $\theta_{\pi_\Omega,i}$.
5:     **end for**
6:     compute $Q_\epsilon$ = upper $\epsilon$ percentile of $\{C(\tau_k)\}_{k=0}^{N_{epi}}$
7:     select sub-set $\mathbb{T} = \{\tau_k | C(\tau_k) \geq Q_\epsilon\}$
8:     $\theta_{\pi_\omega,i+1}, \theta_{\beta_\omega,i+1}, \theta_{\pi_\Omega,i+1} \leftarrow$ PPOCUpdateParams$(\mathbb{T}, \theta_{\pi_\omega,i}, \theta_{\beta_\omega,i}, \theta_{\pi_\Omega,i})$
9: **end for**

---

## Decompositionability of $\mathbb{E}_{C,p}[\max(0, C - v)]$ with Respect to $p$

**Corollary 1** (Decompositionability of $\mathbb{E}_{C,p}[\max(0, C - v)]$ with Respect to $p$)**.**

$$\mathbb{E}_{C,p}[\max(0, C - v)] = \sum_p \mathbb{P}(p)\mathbb{E}_C[\max(0, C - v) \mid p].$$

*Proof.* By letting $\mathbb{P}(C, p)$ be a joint distribution of the loss and the model parameter and $\mathbb{P}(C \mid p)$ be the conditional distribution of the loss, $\mathbb{E}_{C,p}[\max(0, C - v)]$ can be transformed as

$$
\begin{aligned}
\mathbb{E}_{C,p}[\max(0, C - v)] &= \sum_C \sum_p \mathbb{P}(C, p)\max(0, C - v) \\
&= \sum_p \mathbb{P}(p) \sum_C \mathbb{P}(C \mid p)\max(0, C - v) \\
&= \sum_p \mathbb{P}(p)\mathbb{E}_C[\max(0, C - v) \mid p]. \quad (20)
\end{aligned}
$$

□

## Derivation of Option Policy Gradient Theorems for Soft Robust Loss

Here, we derive option policy gradient theorems for soft robust loss $\sum_p \mathbb{P}(p)\mathbb{E}_C[C \mid p]$, where $\mathbb{E}_C[C \mid p]$ is the expected loss on the general class of parameterized MDPs $\langle S, A, \mathbb{C}, \gamma, T_p, \mathbb{P}_0 \rangle^{[10]}$, in which the transition probability is parameterized by $p \in P$. In addition, $\mathbb{P}(p)$ is a distribution of $p$. For convenience, in the latter part, we use $\mathbb{P}(s' \in S \mid s \in S, a \in A, p)$ to refer to the parameterized transition function (i.e., $T_p$). We make the rectangularity assumption on $P$ and $\mathbb{P}(p)$. That is, $P$ is

assumed to be structured as a Cartesian product $\bigotimes_{s \in S} P_s$, and $\mathbb{P}$ is also assumed to be structured as a Cartesian product $\bigotimes_{s \in S} \mathbb{P}_s(p_s \in P_s)$ [11].

To prepare for the derivation, we define functions and variables. First, considering the definition of value functions in Bacon et al. [1], we define value functions in (the general class of) the parameterized MDP:

$$Q_\Omega(s, \omega, p_s) = \sum_a \pi_\omega(a \mid s) Q_\omega(s, \omega, a, p_s), \tag{21}$$

$$Q_\omega(s, \omega, a, p_s) = \mathbb{C}(s, a) + \gamma \sum_{s'} \mathbb{P}(s' \mid s, a, p_s) Q_\beta(s', \omega, p_s), \tag{22}$$

$$Q_\beta(s, \omega, p_s) = (1 - \beta_\omega(s)) Q_\Omega(s, \omega, p_s) + \beta_\omega(s) V_\Omega(s, p_s), \tag{23}$$

$$V_\Omega(s, p_s) = \sum_\omega \pi_\Omega(\omega \mid s) Q_\Omega(s, \omega, p_s). \tag{24}$$

Note that by fixing $p_s$ to a constant value, these value functions can be seen as identical to those in Bacon et al. [1].

Second, for convenience, we define the discounted probabilities to $(s_{t+1}, \omega_{t+1})$ from $(s_t, \omega_t)$:

$$\mathbb{P}_\gamma^{(1)}(s_{t+1}, \omega_{t+1} \mid s_t, \omega_t) = \sum_a \pi_\omega(a \mid s_t) \gamma \sum_{p_{s_t}} \mathbb{P}_{s_t}(p_{s_t}) \mathbb{P}(s_{t+1} \mid s_t, a, p_{s_t})$$

$$\cdot ((1 - \beta_\omega(s_{t+1})) \mathbf{1}(\omega_{t+1} = \omega_t) + \beta_\omega(s_{t+1}) \pi_\Omega(\omega_{t+1} | s_{t+1})), \tag{25}$$

$$\mathbb{P}_\gamma^{(k)}(s_{t+k}, \omega_{t+k} \mid s_t, \omega_t) = \sum_{s_{t+1}} \sum_{\omega_{t+1}} \mathbb{P}_\gamma^{(1)}(s_{t+1}, \omega_{t+1} \mid s_t, \omega_t) \mathbb{P}_\gamma^{(k-1)}(s_{t+k}, \omega_{t+k} \mid s_{t+1}, \omega_{t+1}). \tag{26}$$

Similar discounted probabilities are also introduced in Bacon et al. [1]. Ours (Eq. 25 and Eq. 26) are different from theirs in that the transition function is averaged over the model parameter (i.e., $\sum_{p_{s_t}} \mathbb{P}_{s_t}(p_{s_t}) \mathbb{P}(s_{t+1} \mid s_t, a, p_{s_t}) = \mathbb{E}_{p_s}[\mathbb{P}(s' \mid s, a, p_s)]$).

In addition, by using the rectangularity assumption on $\mathbb{P}(p)$, we introduce the equation which represents the state-wise independence of the model parameter distribution [12]:

$$\sum_{p_{s_t} \in P_{s_t}} \mathbb{P}_{s_t}(p_{s_t}) \sum_{s_{t+1}} \mathbb{P}(s_{t+1} \mid s_t, a_t, p_{s_t}) Q_\Omega(s_{t+1}, \omega_t, p_{s_t})$$

$$= \sum_{s_{t+1}} \left( \sum_{p_{s_t}} \mathbb{P}_{s_t}(p_{s_t}) \mathbb{P}(s_{t+1} \mid s_t, a_t, p_{s_t}) \right) \left( \sum_{p_{s_{t+1}} \in P_{s_{t+1}}} \mathbb{P}_{s_{t+1}}(p_{s_{t+1}}) Q_\Omega(s_{t+1}, \omega_t, p_{s_{t+1}}) \right). \tag{27}$$

Now, we start the derivation of option policy gradient theorems for soft robust loss. We derive the gradient theorem in a similar manner to Bacon et al. [1].

**Theorem 1** (Policy over options gradient theorem for soft robust loss)**.** *Given a set of options with a stochastic policy over options $\pi_\Omega$ that are differentiable in their parameters $\theta_{\pi_\Omega}$, the gradient of the soft robust loss with respect to $\theta_{\pi_\Omega}$ and initial condition $(s_0, \omega_0)$ is:*

$$\sum_{p_s} \mathbb{P}_s(p_s) \frac{\partial Q_\Omega(s, \omega, p_s)}{\partial \theta_{\pi_\Omega}} = \sum_s \overline{d}_\Omega(s \mid s_0, \omega_0) \sum_\omega \frac{\partial \pi_\Omega(\omega|s)}{\partial \theta_{\pi_\Omega}} \overline{Q}_\Omega(s, \omega),$$

*where $\overline{d}_\Omega(s \mid s_0, \omega_0)$ is a discounted weighting of option pairs along trajectories from $(s_0, \omega_0)$: $\overline{d}_\Omega(s \mid s_0, \omega_0) = \sum_{k=0}^\infty \mathbb{P}_\gamma^{(k)}(s \mid s_0, \omega_0)$, and $\overline{Q}_\Omega(s, \omega)$ is the average option value function: $\overline{Q}_\Omega(s, \omega) = \sum_{p_s} \mathbb{P}_s(p_s) Q_\Omega(s, \omega, p_s)$. By letting $\mathbb{E}_{p_s}[\mathbb{P}(s' \mid s, a, p_s)]$ be a transition probability, the trajectories can be regarded as ones generated from the general class of the average parameterized MDPs $\langle S, A, \mathbb{C}, \gamma, \mathbb{E}_{p_s}[\mathbb{P}(s' \mid s, a, p_s)], \mathbb{P}_0 \rangle$.*

*Proof.* The gradient of Eq. 21 and Eq. 23 with respect to $\theta_{\pi_\Omega}$ can be transformed as

$$\frac{\partial Q_\Omega(s,\omega,p_s)}{\partial \theta_{\pi_\Omega}} = \sum_a \pi_\omega(a \mid s) \sum_{s'} \gamma \mathbb{P}(s' \mid s, a, p_s) \frac{\partial Q_\beta(\omega, s', p_s)}{\partial \theta_{\pi_\Omega}}, \tag{28}$$

$$\frac{\partial Q_\beta(s,\omega,p_s)}{\partial \theta_{\pi_\Omega}} = (1 - \beta_\omega(s')) \frac{\partial Q_\Omega(s',\omega,p_s)}{\partial \theta_{\pi_\Omega}} + \beta_\beta(s') \frac{\partial V_\Omega(s',p_s)}{\partial \theta_{\pi_\Omega}}$$

$$= (1 - \beta_\omega(s')) \frac{\partial Q_\Omega(s',\omega,p_s)}{\partial \theta_{\pi_\Omega}} + \beta_\omega(s') \sum_{\omega'} \frac{\partial \pi_\Omega(\omega'|s')}{\partial \theta_{\pi_\Omega}} Q_\Omega(s',\omega',p_s)$$

$$+ \beta_\omega(s') \sum_{\omega'} \pi_\Omega(\omega'|s') \frac{\partial Q_\Omega(s',\omega',p_s)}{\partial \theta_{\pi_\Omega}}$$

$$= \beta_\omega(s') \sum_{\omega'} \frac{\partial \pi_\Omega(\omega'|s')}{\partial \theta_{\pi_\Omega}} Q_\Omega(s',\omega',p_s)$$

$$+ \sum_{\omega'} \left((1 - \beta_\omega(s'))\mathbf{1}(\omega' = \omega) + \beta_\omega(\mathbf{s}')\pi_\Omega(\omega'|\mathbf{s}')\right) \frac{\partial Q_\Omega(s',\omega',p_s)}{\partial \theta_{\pi_\Omega}}. \tag{29}$$

By substituting Eq. 29 into Eq. 28, a recursive expression of the gradient of the value functions is acquired as

$$\frac{\partial Q_\Omega(s,\omega,p_s)}{\partial \theta_{\pi_\Omega}} = \sum_a \pi_\omega(a \mid s) \sum_{s'} \gamma \mathbb{P}(s' \mid s, a, p_s) \beta_\omega(s') \sum_{\omega'} \frac{\partial \pi_\Omega(\omega'|s')}{\partial \theta_{\partial\theta_{\pi_\Omega}}} Q_\Omega(s',\omega',p_s)$$

$$+ \sum_{s'} \sum_{\omega'} \sum_a \pi_\omega(a \mid s) \gamma \mathbb{P}(s' \mid s, a, p_s) \left((1 - \beta_\omega(s'))\mathbf{1}(\omega' = \omega) + \beta_\omega(\mathbf{s}')\pi_\Omega(\omega'|\mathbf{s}')\right) \frac{\partial Q_\Omega(s',\omega',p_s)}{\partial \theta_{\pi_\Omega}}. \tag{30}$$

By using Eq. 27, the gradient of soft robust loss style value functions can also be recursively expressed as

$$\sum_{p_s} \mathbb{P}_s(p_s) \frac{\partial Q_\Omega(s,\omega,p_s)}{\partial \theta_{\pi_\Omega}}$$

$$= \sum_{p_s} \mathbb{P}_s(p_s) \sum_a \pi_\omega(a \mid s) \sum_{s'} \gamma \mathbb{P}(s' \mid s, a, p_s) \beta_\omega(s') \sum_{\omega'} \frac{\partial \pi_\Omega(\omega'|s')}{\partial \theta_{\pi_\Omega}} Q_\Omega(s',\omega',p_s)$$

$$+ \sum_{p_s} \mathbb{P}_s(p_s) \sum_{s'} \sum_{\omega'} \sum_a \pi_\omega(a \mid s) \gamma \mathbb{P}(s' \mid s, a, p_s) \left((1 - \beta_\omega(s'))\mathbf{1}(\omega' = \omega) + \beta_\omega(s')\pi_\Omega(\omega'|s')\right)$$

$$\cdot \frac{\partial Q_\Omega(s',\omega',p_s)}{\partial \theta_{\pi_\Omega}}$$

$$= \sum_a \pi_\omega(a \mid s) \sum_{s'} \gamma \sum_{p_s} \mathbb{P}_s(p_s)\mathbb{P}(s' \mid s, a, p_s) \beta_\omega(s') \sum_{\omega'} \frac{\partial \pi_\Omega(\omega'|s')}{\partial \theta_{\pi_\Omega}} \overline{Q}_\Omega(s',\omega')$$

$$+ \sum_{s'} \sum_{\omega'} \sum_a \pi_\omega(a \mid s) \gamma \sum_{p_s} \mathbb{P}_s(p_s)\mathbb{P}(s' \mid s, a, p_s) \left((1 - \beta_\omega(s'))\mathbf{1}(\omega' = \omega) + \beta_\omega(s')\pi_\Omega(\omega'|s')\right)$$

$$\cdot \sum_{p_{s'}} \mathbb{P}_{s'}(p_{s'}) \frac{\partial Q_\Omega(s',\omega',p_{s'})}{\partial \theta_{\pi_\Omega}}. \tag{31}$$

By using Eq. 25 and Eq. 26, Eq. 31 can be transformed as

$$
\sum_{p_s} \mathbb{P}_s(p_s) \frac{\partial Q_\Omega(s,\omega,p_s)}{\partial \theta_{\pi_\Omega}} = \sum_a \pi_\omega(a \mid s) \sum_{s'} \gamma \sum_{p_s} \mathbb{P}_s(p_s)\mathbb{P}(s' \mid s,a,p_s)\beta_\omega(s') \sum_{\omega'} \frac{\partial \pi_\Omega(\omega'|s')}{\partial \theta_{\pi_\Omega}} \overline{Q}_\Omega(s',\omega')
$$

$$
+ \sum_{s'} \sum_{\omega'} \mathbb{P}_\gamma^{(1)}(s',\omega' \mid s,\omega) \sum_{p_{s'}} \mathbb{P}_{s'}(p_{s'}) \frac{\partial Q_\Omega(s',\omega',p_{s'})}{\partial \theta_{\pi_\Omega}}
$$

$$
= \sum_{k=0}^{\infty} \sum_{s''} \underbrace{\sum_{s'} \sum_{\omega'} \mathbb{P}_\gamma^{(k)}(s',\omega' \mid s,\omega) \sum_a \pi_{\omega'}(a \mid s')\gamma \sum_{p_{s'}} \mathbb{P}_{s'}(p_{s'})\mathbb{P}(s'' \mid s',a,p_{s'})\beta_{\omega'}(s'')}_{\mathbb{P}_\gamma^{(k)}(s''|s,\omega)}
$$

$$
\cdot \sum_{\omega''} \frac{\partial \pi_\Omega(\omega''|s'')}{\partial \theta_{\pi_\Omega}} \overline{Q}_\Omega(s'',\omega''). \tag{32}
$$

The gradient of the expected discounted soft robust loss with respect to $\theta_{\pi_\Omega}$ is then

$$
\begin{aligned}
\sum_{p_s} \mathbb{P}_s(p_s) \frac{\partial Q_\Omega(s,\omega,p_s)}{\partial \theta_{\pi_\Omega}} &= \sum_{k=0}^{\infty} \sum_s \mathbb{P}_\gamma^{(k)}(s \mid s_0,\omega_0) \sum_\omega \frac{\partial \pi_\Omega(\omega|s)}{\partial \theta_{\pi_\Omega}} \overline{Q}_\Omega(s,\omega) \\
&= \sum_s \overline{d}_\Omega(s \mid s_0,\omega_0) \sum_\omega \frac{\partial \pi_\Omega(\omega|s)}{\partial \theta_{\pi_\Omega}} \overline{Q}_\Omega(s,\omega). \tag{33}
\end{aligned}
$$

□

**Theorem 2** (Intra-option policy gradient theorem for soft robust loss). *Given a set of options with stochastic intra-option policies $\pi_\omega$ that are differentiable in their parameters $\theta_{\pi_\omega}$, the gradient of soft robust loss with respect to $\theta_{\pi_\omega}$ and initial condition $(s_0,\omega_0)$ is:*

$$
\sum_{p_s} \mathbb{P}_s(p_s) \frac{\partial Q_\Omega(s,\omega,p_s)}{\partial \theta_{\pi_\omega}} = \sum_s \sum_\omega \overline{d}_\Omega(s,\omega \mid s_0,\omega_0) \sum_a \frac{\partial \pi_\omega(a \mid s)}{\partial \theta_{\pi_\omega}} \overline{Q}_\omega(s,\omega,a), \tag{34}
$$

*where $\overline{d}_\Omega(s,\omega \mid s_0,\omega_0)$ is a discounted weighting of state option pairs along trajectories from $(s_0,\omega_0)$: $\overline{d}_\Omega(s,\omega \mid s_0,\omega_0) = \sum_{t=0}^{\infty} \gamma^t \mathbb{P}_\gamma^{(k)}(s,\omega \mid s_0,\omega_0)$, and $\overline{Q}_\omega(s,\omega,a)$ is the average value function of actions in the context of a state-option pair over the model parameter distribution: $\overline{Q}_\omega(s,\omega,a) = \sum_{p_s} \mathbb{P}_s(p_s)Q_\omega(s,\omega,a,p_s)$. By letting $\mathbb{E}_{p_s}\left[\mathbb{P}(s' \mid s,a,p_s)\right]$ be a transition probability, the trajectories can be regarded as ones generated from the general class of the average parameterized MDPs $\langle S, A, \mathbb{C}, \gamma, \mathbb{E}_{p_s}\left[\mathbb{P}(s' \mid s,a,p_s)\right], \mathbb{P}_0 \rangle$.*

*Proof.* The gradient of Eq. 22 with respect to $\theta_{\pi_\omega}$ can be recursively written as

$$
\begin{aligned}
\frac{\partial Q_\Omega(s,\omega,p_s)}{\partial \theta_{\pi_\omega}} &= \sum_a \frac{\partial \pi_\omega(a \mid s)}{\partial \theta_{\pi_\omega}} Q_\omega(s,\omega,a,p_s) \\
&+ \sum_a \pi_\omega(a \mid s)\gamma \sum_{s'} \mathbb{P}(s \mid s,a,p_s) \sum_{\omega'} \left(\beta_\omega(s')\pi_\Omega(\omega' \mid s') + (1 - \beta_\omega(s')\mathbf{1}(\omega' = \omega))\right) \\
&\cdot \frac{\partial Q_\Omega(s',\omega',p_s)}{\partial \theta_{\pi_\omega}}. \tag{35}
\end{aligned}
$$

By using Eq. 27, the gradient of the soft robust style value function can be recursively expressed as

$$
\begin{aligned}
\sum_{p_s} \mathbb{P}_s(p_s) \frac{\partial Q_\Omega(s,\omega,p_s)}{\partial \theta_{\pi_\omega}} &= \sum_a \frac{\partial \pi_\omega(a \mid s)}{\partial \theta_{\pi_\omega}} \sum_{p_s} \mathbb{P}_s(p_s)Q_\omega(s,\omega,a,p_s) \\
&+ \sum_a \pi_\omega(a \mid s) \sum_{s'} \gamma \sum_{p_s} \mathbb{P}_s(p_s)\mathbb{P}(s \mid s,a,p_s) \sum_{\omega'} \left(\beta_\omega(s')\pi_\Omega(\omega' \mid s') + (1 - \beta_\omega(s')\mathbf{1}(\omega' = \omega))\right) \\
&\cdot \sum_{p_{s'}} \mathbb{P}_{s'}(p_{s'}) \frac{\partial Q_\Omega(s',\omega',p_{s'})}{\partial \theta_{\pi_\omega}}. \tag{36}
\end{aligned}
$$

By using Eq. 25 and Eq. 26, Eq. 36 can be transformed as

$$
\begin{aligned}
\sum_{p_s} \mathbb{P}_s(p_s) \frac{\partial Q_\Omega(s, \omega, p_s)}{\partial \theta_{\pi_\omega}} &= \sum_a \frac{\partial \pi_\omega(a \mid s)}{\partial \theta_{\pi_\omega}} \sum_{p_s} \mathbb{P}_s(p_s) Q_\omega(s, \omega, a, p_s) \\
&\quad + \sum_{s'} \sum_{\omega'} \mathbb{P}_\gamma^{(1)}(s', \omega' \mid s, \omega) \sum_{p_{s'}} \mathbb{P}_{s'}(p_{s'}) \frac{\partial Q_\Omega(s', \omega', p_{s'})}{\partial \theta_{\pi_\omega}} \\
&= \sum_{k=0}^{\infty} \sum_{s'} \sum_{\omega'} \mathbb{P}_\gamma^{(k)}(s', \omega' \mid s, \omega) \sum_a \frac{\partial \pi_\omega(a \mid s')}{\partial \theta_{\pi_\omega}} \\
&\quad \cdot \sum_{p_{s'}} \mathbb{P}_{s'}(p_{s'}) Q_\omega(s', \omega', a, p_{s'}) \\
&= \sum_{k=0}^{\infty} \sum_{s'} \sum_{\omega'} \mathbb{P}_\gamma^{(k)}(s', \omega' \mid s, \omega) \sum_a \frac{\partial \pi_\omega(a \mid s')}{\partial \theta_{\pi_\omega}} \overline{Q}_\omega(s', \omega', a). \quad (37)
\end{aligned}
$$

The gradient of the expected discounted soft robust loss with respect to $\theta_{\pi_\omega}$ is then

$$
\begin{aligned}
\sum_{p_s} \mathbb{P}_s(p_s) \frac{\partial Q_\Omega(s, \omega, p_s)}{\partial \theta_{\pi_\omega}} &= \sum_{k=0}^{\infty} \sum_s \sum_\omega \mathbb{P}_\gamma^{(k)}(s, \omega \mid s_0, \omega_0) \sum_a \frac{\partial \pi_\omega(a \mid s)}{\partial \theta_{\pi_\omega}} \overline{Q}_\omega(s, \omega, a) \\
&= \sum_s \sum_\omega \overline{d}_\Omega(s, \omega \mid s_0, \omega_0) \sum_a \frac{\partial \pi_\omega(a \mid s)}{\partial \theta_{\pi_\omega}} \overline{Q}_\omega(s, \omega, a). \quad (38)
\end{aligned}
$$

$\square$

**Theorem 3** (Termination function gradient theorem for soft robust loss). *Given a set of options with stochastic termination functions $\beta_\omega$ that are differentiable in their parameters $\theta_{\beta_\omega}$, the gradient of the expected soft robust loss with respect $\theta_{\beta_\omega}$ is:*

$$
\sum_{p_s} \mathbb{P}_s(p_s) \frac{\partial Q_\Omega(s, \omega, p_s)}{\partial \theta_{\beta_\omega}} = \sum_s \sum_\omega \overline{d}_\Omega(s, \omega \mid s_0, \omega_0) \frac{\partial \beta_\omega(s)}{\partial \theta_{\beta_\omega}} \left( \overline{V}_\Omega(s) - \overline{Q}_\Omega(s, \omega) \right), \quad (39)
$$

*where $\overline{d}_\Omega(s, \omega \mid s_0, \omega_0)$ is a discounted weighting of state option pairs along trajectories from $(s_0, \omega_0)$: $\overline{d}_\Omega(s, \omega \mid s_0, \omega_0) = \sum_{t=0}^{\infty} \gamma^t \mathbb{P}_\gamma^{(k)}(s, \omega \mid s_0, \omega_0)$. In addition, $\overline{Q}_\Omega(s, \omega)$ is the value function of options in the context of a state, which is averaged over the model parameter distribution: $\overline{Q}_\Omega(s, \omega) = \sum_{p_s} \mathbb{P}_s(p_s) Q_\Omega(s, \omega, p_s)$, and $\overline{V}_\Omega(s)$ is the value function averaged over the model parameter distribution: $\overline{V}_\Omega(s) = \sum_{p_s} \mathbb{P}_s(p_s) \overline{V}_\Omega(s, p_s)$. By letting $\mathbb{E}_{p_s} [\mathbb{P}(s' \mid s, a, p_s)]$ as a transition probability, the trajectories can be regarded as ones generated from the general class of the average parameterized MDPs $\langle S, A, \mathbb{C}, \gamma, \mathbb{E}_{p_s} [\mathbb{P}(s' \mid s, a, p_s)], \mathbb{P}_0 \rangle$.*

*Proof.* The gradient of Eq. 24 with respect to $\theta_{\beta_\omega}$ can be written as follows:

$$
\frac{\partial Q_\Omega(s, \omega, p_s)}{\partial \theta_{\beta_\omega}} = \sum_a \pi_\omega(a \mid s) \sum_{s'} \gamma \mathbb{P}(s' \mid s, a, p_s) \frac{\partial Q_\beta(\omega, s', p_s)}{\partial \theta_{\beta_\omega}}. \quad (40)
$$

In addition, the gradient of Eq. 23 with respect to $\theta_{\beta_\omega}$ can be written as

$$
\begin{aligned}
\frac{\partial Q_\beta(\omega, s, p_s)}{\partial \theta_{\beta_\omega}} &= -\frac{\partial \beta_\omega(s)}{\partial \theta_{\beta_\omega}} Q_\Omega(s, \omega, p_s) + (1 - \beta_\omega(s)) \frac{\partial Q_\Omega(s, \omega, p)}{\partial \theta_{\beta_\omega}} + \frac{\partial \beta_\omega(s)}{\partial \theta_{\beta_\omega}} V_\Omega(s, p) \\
&\quad + \beta_\omega(s) \frac{\partial V_\Omega(s, p_s)}{\partial \theta_{\beta_\omega}} \\
&= \frac{\partial \beta_\omega(s')}{\partial \theta_{\beta_\omega}} (V_\Omega(s, p_s) - Q_\Omega(s, \omega, p_s)) \\
&\quad + \sum_{\omega'} \left( (1 - \beta_\omega(s)) \mathbf{1}(\omega' = \omega) + \beta_\omega(s) \pi_\Omega(\omega' \mid s) \right) \frac{\partial Q_\Omega(s, \omega', p_s)}{\partial \theta_{\beta_\omega}}. \quad (41)
\end{aligned}
$$

By substituting Eq. 41 into Eq. 40, a recursive expression of the gradient can be written as

$$
\frac{\partial Q_\Omega(s, \omega, p_s)}{\partial \theta_{\beta_\omega}} = \sum_a \pi_\omega(a \mid s) \sum_{s'} \gamma \mathbb{P}_s(s' \mid s, a, p_s) \frac{\partial \beta_\omega(s')}{\partial \theta_{\beta_\omega}} \left( V_\Omega(s', p_s) - Q_\Omega(s', \omega, p_s) \right)
$$

$$
+ \sum_a \pi_\omega(a \mid s) \sum_{s'} \gamma \mathbb{P}(s' \mid s, a, p_s) \sum_{\omega'} \left( (1 - \beta_\omega(s'))\mathbf{1}(\omega' = \omega) + \beta_\omega(s')\pi_\Omega(\omega' \mid s') \right) \frac{\partial Q_\Omega(s', \omega', p_s)}{\partial \theta_{\beta_\omega}}.
$$
(42)

By using Eq. 27, the gradient of soft robust loss style value functions can also be recursively expressed as

$$
\sum_{p_s} \mathbb{P}_s(p_s) \frac{\partial Q_\Omega(s, \omega, p_s)}{\partial \theta_{\beta_\omega}} = \sum_a \pi_\omega(a \mid s) \sum_{s'} \gamma \sum_{p_s} \mathbb{P}_s(p_s)\mathbb{P}(s' \mid s, a, p_s) \frac{\partial \beta_\omega(s')}{\partial \theta_{\beta_\omega}} \left( \overline{V}_\Omega(s') - \overline{Q}_\Omega(s', \omega) \right)
$$

$$
+ \sum_a \pi_\omega(a \mid s) \sum_{s'} \gamma \sum_{p_s} \mathbb{P}_s(p_s)\mathbb{P}_{s'}(s' \mid s, a, p_s) \sum_{\omega'} \left( (1 - \beta_\omega(s'))\mathbf{1}(\omega' = \omega) + \beta_\omega(s')\pi_\Omega(\omega' \mid s') \right)
$$

$$
\cdot \sum_{p_{s'}} \mathbb{P}_{s'}(p_{s'}) \frac{\partial Q_\Omega(s', \omega', p_{s'})}{\partial \theta_{\beta_\omega}}.
$$
(43)

By using Eq. 26, Eq. 43 can be transformed as

$$
\sum_{p_s} \mathbb{P}_s(p_s) \frac{\partial Q_\Omega(s, \omega, p_s)}{\partial \theta_{\beta_\omega}} = \sum_a \pi_\omega(a \mid s) \sum_{s'} \gamma \sum_{p_s} \mathbb{P}(p_s)\mathbb{P}_s(s' \mid s, a, p_s) \frac{\partial \beta_\omega(s')}{\partial \theta_{\beta_\omega}} \left( \overline{V}_\Omega(s') - \overline{Q}_\Omega(s', \omega) \right)
$$

$$
+ \sum_{s'} \sum_{\omega'} \mathbb{P}_\gamma^{(1)}(s', \omega' \mid s, \omega) \sum_{p_{s'}} \mathbb{P}(p_{s'}) \frac{\partial Q_\Omega(s', \omega', p_{s'})}{\partial \theta_{\beta_\omega}}
$$

$$
= \sum_{k=0}^{\infty} \sum_{s'} \sum_{\omega'} \mathbb{P}_\gamma^{(k)}(s', \omega' \mid s, \omega) \sum_a \pi_\omega(a \mid s') \sum_{s''} \gamma \sum_{p_{s'}} \mathbb{P}_s(p_{s'})\mathbb{P}(s'' \mid s', a, p_{s'})
$$

$$
\cdot \frac{\partial \beta_{\omega'}(s'')}{\partial \theta_{\beta_\omega}} \left( \overline{V}_\Omega(s'') - \overline{Q}_\Omega(s'', \omega') \right)
$$

$$
= \sum_{k=0}^{\infty} \sum_{s''} \sum_{\omega'} \underbrace{\sum_{s'} \mathbb{P}_\gamma^{(k)}(s', \omega' \mid s, \omega) \cdot \sum_a \pi_\omega(a \mid s')\gamma \sum_{p_{s'}} \mathbb{P}_s(p_{s'})\mathbb{P}(s'' \mid s', a, p_{s'})}_{\mathbb{P}_\gamma^{(k)}(s'', \omega' \mid s, \omega)}
$$

$$
\cdot \frac{\partial \beta_{\omega'}(s'')}{\partial \theta_{\beta_\omega}} \left( \overline{V}_\Omega(s'') - \overline{Q}_\Omega(s'', \omega') \right).
$$
(44)

The gradient of the expected discounted soft robust loss with respect to $\theta_{\beta_\omega}$ is then

$$
\begin{aligned}
\sum_{p_s} \mathbb{P}_s(p_s) \frac{\partial Q_\Omega(s, \omega, p_s)}{\partial \theta_{\beta_\omega}} &= \sum_{k=0}^{\infty} \sum_s \sum_\omega \mathbb{P}_\gamma^{(k)}(s, \omega \mid s_0, \omega_0) \frac{\partial \beta_\omega(s)}{\partial \theta_{\beta_\omega}} \left( \overline{V}_\Omega(s) - \overline{Q}_\Omega(s, \omega) \right) \\
&= \sum_s \sum_\omega \overline{d}_\Omega(s, \omega \mid s_0, \omega_0) \frac{\partial \beta_\omega(s)}{\partial \theta_{\beta_\omega}} \left( \overline{V}_\Omega(s) - \overline{Q}_\Omega(s, \omega) \right).
\end{aligned}
$$
(45)

$\square$

In addition, we derive a corollary for the derivation of the gradient of $\lambda$ and $\nu$.

**Corollary 2** (Relation between the soft robust loss over parameterized MDPs and the loss on an average parameterized MDP)**.**

$$
\mathbb{E}_{C,p}[C] = \mathbb{E}_C[C \mid \overline{p}].
$$
(46)

*Proof.* For proof, considering the definition of value functions in Bacon et al. [1], we define value functions on (the general class of) an average parameterized MDP $\langle S, A, \mathbb{C}, \gamma, \mathbb{E}_{p_s}\left[\mathbb{P}(s' \mid s, a, p_s)\right], \mathbb{P}_0 \rangle$:

$$\overline{Q}_\Omega(s, \omega) = \sum_a \pi_\omega(a \mid s)\overline{Q}_\omega(s, \omega, a), \tag{47}$$

$$\overline{Q}_\omega(s, \omega, a) = \mathbb{C}(s, a) + \gamma \sum_{s'} \mathbb{E}_{p_s}\left[\mathbb{P}(s' \mid s, a, p_s)\right]\overline{Q}_\beta(s', \omega), \tag{48}$$

$$\overline{Q}_\beta(s, \omega) = (1 - \beta_\omega(s))\overline{Q}_\Omega(s, \omega) + \beta_\omega(s)\overline{V}_\Omega(s), \tag{49}$$

$$\overline{V}_\Omega(s) = \sum_\omega \pi_\Omega(\omega \mid s)\overline{Q}_\Omega(s, \omega). \tag{50}$$

Note that, by regarding $\mathbb{E}_{p_s}\left[\mathbb{P}(s' \mid s, a, p_s)\right]$ as a transition functions, the definition of value functions become identical to those in Bacon et al. [1].

By using Eq. 50, the loss at the average parameterized MDP can be written as

$$\mathbb{E}_C\left[C \mid \overline{p}\right] = \overline{V}_\Omega(s_0). \tag{51}$$

In addition, with Eq. 24, the soft robust loss can be written as

$$\mathbb{E}_{C,p}\left[C\right] = \sum_{p_{s_0}} \mathbb{P}_{s_0}(p_{s_0})V_\Omega(s_0, p_{s_0}). \tag{52}$$

In the following part, we prove $\sum_{p_{s_0}} \mathbb{P}_{s_0}(p_{s_0})V_\Omega(s_0, p_{s_0}) = \overline{V}_\Omega(s_0)$ by backward induction.
**The case at the terminal state $T$:**

$$\sum_{p_{s_T}} \mathbb{P}_{s_T}(p_{s_T})V_\Omega(s_T, p_{s_T}) = \sum_{p_{s_T}} \mathbb{P}_{s_T}(p_{s_T}) \sum_\omega \pi_\Omega(\omega \mid s_T)Q_\Omega(s_T, \omega, p_{s_T})$$

$$= \sum_{p_{s_T}} \mathbb{P}_{s_T}(p_{s_T}) \sum_\omega \pi_\Omega(\omega \mid s_T) \sum_a \pi_\omega(a \mid s_T)\mathbb{C}(s_T, a)$$

$$= \sum_\omega \pi_\Omega(\omega \mid s_T) \sum_a \pi_\omega(a \mid s_T)\mathbb{C}(s_T, a). \tag{53}$$

$$\overline{V}_\Omega(s_T) = \sum_\omega \pi_\Omega(\omega \mid s_T)\overline{Q}_\Omega(s_T, \omega)$$

$$= \sum_\omega \pi_\Omega(\omega \mid s_T) \sum_a \pi_\omega(a \mid s_T)\mathbb{C}(s_T, a). \tag{54}$$

From Eq. 53 and Eq. 54, it is clear that $\sum_{p_{s_T}} \mathbb{P}_{s_T}(p_{s_T})V_\Omega(s_T, p) = \overline{V}_\Omega(s_T)$, and $\sum_{p_{s_T}} \mathbb{P}_{s_T}(p_{s_T})Q_\Omega(s_T, \omega, p_{s_T}) = \overline{Q}_\Omega(s_T, \omega)$.

**The case at the state in $t-1$, while assuming that $\sum_{p_{s_t}} \mathbb{P}_{s_t}(p_{s_t})V_\Omega(s_t, p_{s_t}) = \overline{V}_\Omega(s_t)$ and $\sum_{p_{s_t}} \mathbb{P}_{s_t}(p_{s_t})Q_\Omega(s_t, \omega, p_{s_t}) = \overline{Q}_\Omega(s_t, \omega)$:**
By substituting Eq.21, Eq.22, and Eq.23, $V_\Omega(s_{t-1}, p)$ can be expanded:

$$V_\Omega(s_{t-1}, p_{s_{t-1}}) = \sum_\omega \pi_\Omega(\omega \mid s_{t-1})Q_\Omega(s_{t-1}, \omega, p_{s_{t-1}})$$

$$= \sum_\omega \pi_\Omega(\omega \mid s_{t-1}) \sum_a \pi_\omega(a \mid s_{t-1})$$

$$\cdot \left(\mathbb{C}(s_{t-1}, a) + \gamma \sum_{s_t} \mathbb{P}(s_t \mid s_{t-1}, a, p_{s_{t-1}})\left((1 - \beta_\omega(s_t))Q_\Omega(s_t, \omega, p_{s_{t-1}}) + \beta_\omega(s_t)V_\Omega(s_t, p_{s_{t-1}})\right)\right). \tag{55}$$

By using the rectangularity assumption on $\mathbb{P}(p)$, the expectation of Eq. 55 can be transformed into

$$\sum_{p_{s_{t-1}}} \mathbb{P}_{s_{t-1}}(p_{s_{t-1}})V_\Omega(s_{t-1}, p_{t-1}) = \sum_{p_{s_{t-1}}} \mathbb{P}_{s_{t-1}}(p_{s_{t-1}}) \sum_\omega \pi_\Omega(\omega \mid s_{t-1})Q_\Omega(s_{t-1}, \omega, p_{t-1})$$

$$= \sum_\omega \pi_\Omega(\omega \mid s_{t-1}) \sum_a \pi_\omega(a \mid s_{t-1})$$

$$\cdot \left( \mathbb{C}(s_{t-1}, a) + \gamma \sum_{s_t} \sum_{p_{s_{t-1}}} \mathbb{P}_{s_{t-1}}(p_{s_{t-1}})\mathbb{P}(s_t \mid s_{t-1}, a, p_{s_{t-1}}) \begin{pmatrix} (1-\beta_\omega(s_t))\sum_{p_t}\mathbb{P}_{s_t}(p_{s_t})Q_\Omega(s_t, \omega, p_{s_t}) \\ +\beta_\omega(s_t)\sum_{p_t}\mathbb{P}_{s_t}(p_{s_t})V_\Omega(s_t, p_{s_t}) \end{pmatrix} \right).$$
(56)

By applying the assumption on value functions at $t$ and letting $\sum_{p_{s_{t-1}}} \mathbb{P}_{s_{t-1}}(p_{s_{t-1}})\mathbb{P}(s_t \mid s, a, p_{s_{t-1}})$ be $\mathbb{E}_{p_{s_{t-1}}}[\mathbb{P}(s_t \mid s_{t-1}, a, p_{s_{t-1}})]$, Eq. 56 can be further transformed into

$$\sum_\omega \pi_\Omega(\omega \mid s_{t-1}) \sum_a \pi_\omega(a \mid s_{t-1})$$

$$\cdot \underbrace{\left( \mathbb{C}(s_{t-1}, a) + \gamma \sum_{s_t} \mathbb{E}_{s_{t-1}}[\mathbb{P}(s_t \mid s_{t-1}, a, p_{s_{t-1}})] \underbrace{\left( (1-\beta_\omega(s_t))\overline{Q}_\Omega(s_t, \omega) + \beta_\omega(s_t)\overline{V}_\Omega(s_t) \right)}_{\overline{Q}_\beta(s_t,\omega)} \right)}_{\overline{Q}_\omega(s_{t-1},\omega,a)}$$

$$= \sum_\omega \pi_\Omega(\omega \mid s_{t-1})\overline{Q}_\Omega(s_{t-1}, \omega) = \overline{V}_\Omega(s_{t-1}).$$
(57)

From Eq. 56 and Eq. 57, it is clear that $\sum_{p_{s_{t-1}}} \mathbb{P}_{s_{t-1}}(p_{s_{t-1}})V_\Omega(s_{t-1}, p_{s_{t-1}}) = \overline{V}_\Omega(s_{t-1})$, and $\sum_{p_{s_{t-1}}} \mathbb{P}_{s_{t-1}}(p_{s_{t-1}})Q_\Omega(s_{t-1}, \omega, p_{s_{t-1}}) = \overline{Q}_\Omega(s_{t-1}, \omega)$.

Finally, by letting $t = 1$, we obtain $\sum_{p_{s_0}} \mathbb{P}_{s_0}(p_{s_0})V_\Omega(s_0, p_{s_0}) = \overline{V}_\Omega(s_0)$.

$\square$

## Parameter Distributions for Our Experiment in Section 4

Table 1: Parameter distributions for our experiment. $\mu$ and $\sigma$ are the mean and standard deviation for a Gaussian distribution, respectively. In addition, high and low are the upper and lower bounds of model parameters, respectively.

| Half-Cheetah | $\mu$ | $\sigma$ | low | high |
|---|---|---|---|---|
| torso mass | 7.0 | 3.0 | 1.0 | 13.0 |
| ground friction | 1.6 | 0.8 | 0.1 | 3.1 |
| joint damping | 6.0 | 2.5 | 1.0 | 11.0 |
| **Walker2D** | $\mu$ | $\sigma$ | low | high |
| torso mass | 6.0 | 1.5 | 3.0 | 9.0 |
| ground friction | 1.9 | 0.4 | 0.9 | 2.9 |
| **HopperIceBlock** | $\mu$ | $\sigma$ | low | high |
| ground friction | 1.05 | 0.475 | 0.1 | 2.0 |

Table 2: Parameter distributions for our experiment. *value* is a possible model parameter value and $\mathbb{P}(value)$ is the probability for each of values.

| Half-Cheetah | value | $\mathbb{P}(value)$ |
|---|---|---|
| Torso mass | 1.0 | 0.9 |
| | 13.0 | 0.1 |
| Ground friction | 0.1 | 0.9 |
| | 3.1 | 0.1 |
| Joint damping | 1.0 | 0.9 |
| | 11.0 | 0.1 |
| **Walker2D** | value | $\mathbb{P}(value)$ |
| Torso mass | 3.0 | 0.9 |
| | 9.0 | 0.1 |
| Ground friction | 0.9 | 0.9 |
| | 2.9 | 0.1 |
| **HopperIceBlock** | value | $\mathbb{P}(value)$ |
| Ground friction | 0.1 | 0.1 |
| | 2.0 | 0.9 |

## The Values of $\zeta$ and The Numbers of Successful Learning Trials in Our Experiment in Section 4

Table 3: The values of $\zeta$ for OC3 in Section 4.

| Half-Cheetah-cont | Walker2D-cont | HopperIceBlock-cont | Half-Cheetah-disc | Walker2D-disc | HopperIceBlock-disc |
|---|---|---|---|---|---|
| $-106.1$ | $-1169.9$ | $-441.3$ | $214.9$ | $-905.1$ | $-331.7$ |

Table 4: The numbers of successful learning trials in which the options produced by OC3 satisfy the CVaR constraints. The values in the brackets are the total numbers of learning trials in each environment.

| Half-Cheetah-cont | Walker2D-cont | HopperIceBlock-cont | Half-Cheetah-disc | Walker2D-disc | HopperIceBlock-disc |
|---|---|---|---|---|---|
| 36 (36) | 35 (36) | 35 (36) | 36 (36) | 35 (36) | 36 (36) |

## The Performance of Methods in Environments with Different Model Parameter Values

In Section 4, we elucidated the answers to questions from the viewpoint of the average-case loss (i.e., Eq. 1) and the worst-case loss (i.e., CVaR). Since the average-case loss and the worst-case loss are summarized scores, readers may want to know about these scores in more detail. For this, we evaluate the options, which are learned in Section 4, by varying model parameter values. The performance (cumulative rewards[13]) of each method is shown in Figures 4 and 5. In the following paragraphs, we rediscuss Q2 and Q3 on the basis of these results.

Regarding **Q2**, we compare the performances of OC3 and SoftRobust, in the environments with the worst-case model parameter value. From Figures 4 and 5, we can see that OC3 performs almost the same or even better than SoftRobust, in the environments with the worst-case model parameter value. For example, in Walker2D with a continuous model parameter distribution (b in Figure 4), the minimum performance of OC3 is at Ground friction = 9.0. This minimum performance is significantly better than that of SoftRobust (the performance at Torso mass = 9). For another example, in HopperIceBlock with the discrete distribution (c in Figure 5), minimum performances of all the methods are at Ground friction = 0.1. Here, the minimum performance of OC3 is significantly higher than that of SoftRobust.

Examples of the motion trajectories (Figure 6) indicate that SoftRobust tends to ignore rare cases in learning options, whereas OC3 considers them. SoftRobust produces option policies that cause a hopper to run with its torso overly bent forward. Although the policies enable the hopper to easily jump over the box in the ordinary case (ground friction = 2.0), they cause the hopper to slip and fall in the rare case (ground friction = 0.1). This illustrates that SoftRobust does not sufficiently consider the rare case while learning options. On the other hand, OC3 produces option policies that lead the hopper to move by waggling its foot while keeping its torso vertical against the ground. In the ordinary case, the hopper hops up, lands on the box, and then passes through the box by slipping on it. In the rare case, the hopper stops its movement when it reaches the place near the box, without hopping onto it. This behaviour prevents the hopper from slipping and falling in the rare case. Further examples are shown in the video at the following link: `https://drive.google.com/open?id=1DRmIaK5VomCey70rKD_5DgX2Jm_1rFlo`

Regarding **Q3**, we compare the performance of OC3 with that of the worst-case methods (WorstCase and EOOpt-0.1) in the ordinary case. For performance in environments with a continuous distribution, we compare their performances on model parameter values that appeared frequently in the learning phase (i.e., performance around the center point in Figure 4). From this viewpoint, we can see that OC3 performs significantly better than the worst-case methods. For example, in HopperIceBlock, OC3 performs significantly better around Ground friction = 1.05 than WorstCase and EOOpt-0.1. For performance in environments with discrete distributions, we compare the performances of the methods in the frequent cases. From the comparison, we can see that OC3 perform better than the

(a) Half-Cheetah

(b) Walker2D

(c) HopperIceBlock

Figure 4: Comparison of methods in environments with different model parameter values. In each figure, the vertical axis represents performance (**the negative of the loss**) of each method and the horizontal axis represents the model parameter value. Each score is averaged over 36 trials with different initial random seeds, and the 95% confidence interval is attached to the score. In option learning, the model parameter values are probabilistically generated by continuous distributions shown in Table 1. Therefore, in each figure, the value of the model parameter which is closer to the center point appears more frequently.

worst-case methods in all cases. For example, in HopperIceBlock, OC3 performs significantly better at Ground friction = 2 than WorstCase and EOOpt-0.1.

Additionally, we conduct an experimental evaluation with the condition that model parameter distribution and parameter value ranges in the test phase are different from those in the training phase. In the training phase, the option policies [14] are learned in environments with continuous model parameter distribution (Half-Cheetah-cont, Walker2D-cont, and HopperIceBlock-cont), which is truncated Gaussian distribution with the parameters shown in Table 1. In the test phase, the model parameter distribution is changed to uniform distribution. The range of the distribution is determined as $[(high + low)/2 - scale factor \cdot (high - low)/2, (high + low)/2 + scale factor \cdot (high - low)/2]$, where low and high are ones in Table 1 and scale factor is non-negative real number to scale the value range [15]. We evaluate the learned options with varying the scale factor. The results (Figrue 7) shows that the performance of OC3 is better than that of SoftRobust when the scale factor is large (i.e., when the model parameter distibution is highly uncertain).

(a) Half-Cheetah

(b) Walker2D

(c) HopperIceBlock

Figure 5: Comparison of methods in environments with different model parameter values. In each figure, the vertical axis represents the performance (**the negative of the loss**) of each method and the horizontal axis represents the model parameter value. Each score is averaged over 36 trials with different initial random seeds, and the 95% confidence interval is attached to the score. In option learning, the model parameter values are probabilistically generated by discrete distributions shown in Table 2. For Half-Cheetah and Walker2D, the model parameter values on the left side of each figure appear more frequently while learning options. In addition, for HopperIceBlock, the model parameter values on the right side of the figure appears more frequently while learning options.

(a) SoftRobust at friction= 0.1

(b) SoftRobust at friction= 2.0

(c) OC3 at friction= 0.1

(d) OC3 at friction= 2.0

Figure 6: Example motion trajectories generated by SoftRobust and OC3 in the HopperIceBlock environment with the discrete model parameter distribution.

(a) Half-Cheetah-cont

(b) Walker2D-cont

(c) HopperIceBlock-cont

Figure 7: The performance of SoftRobust and OC3 in environments with uniform model parameter distribution. In each figure, the vertical axis represents the performance (**the negative of the loss**) of each method and the horizontal axis represents the scale factor for the range of the uniform distribution. Each score is averaged over 36 trials with different initial random seeds, and the 95% confidence interval is attached to the score.

## Footnotes

[10] For derivation on the general case, we consider the general class of parameterized MDPs, to which the parameterized MDP in Section 2 and the augmented parameterized MDP in Section 3.1 belong.

[11] More explicitly, $P$ and $\mathbb{P}(p)$ are defined as $\bigotimes_{t0}^\infty \bigotimes_{s \in S} P_{s,t}$ and $\bigotimes_{t0}^\infty \bigotimes_{s \in S} \mathbb{P}_{s,t}(p_{s,t} \in P_{s,t})$, where $P_{s,t} = P_s$ and $P_{s,t}(p_s \in P_{s,t}) = P_s(p_s \in P_s)$, respectively.

[12] This state-wise independence is essentially the same as that introduced in the proof of Proposition 3.2 in Derman et al. [9].

[13]This is equal to the negative of the loss.

[14]Here, we compare ones learned by OC3 and SoftRobust.

[15]If the model parameter values sampled from the distribution are invalid (i.e., negative or zero), these are replaced by the value 0.0001.