[Reviews · NeurIPS 2019]

Reviewer 1



Originality: The work puts together ideas and formulations from various areas. I believe that this work is an interesting extension of the seminal work on CVaR-based RL by Chow and Ghavamzadeh, but I would not consider it as particularly novel. It mainly makes use of the Chow and Ghavamzadeh machinery and proposes an interesting but relatively straightforward extension. 2. Quality: I believe the work is technically correct, even though I was not able to go through all details. Prior works are properly cited and the authors provide proofs to their claims. 3. Clarity: The paper is based on a number of previous works, so it is not easily accessible to an audience without the necessary background. But I think it is clearly written and knowledgeable readers should be able to follow the paper. 4. Significance: Learning options is quite important in the RL context, and robust options have been studied recently [Mankowitz et al.]. Even though I do not feel that the present work advances the theory of robust options in some major way, it nevertheless provides an interesting framework for robust options that additionally incorporate the CVaR concept. Some practitioners may find such ideas useful. UPDATE I have read the rebuttal and the other reviews, and I appreciate the authors' response to the points I raised. My overall feeling is that this is an interesting work in the field of reinforcement learning with robust options, and the experiments (old and new) look promising. With that said, all reviewers agreed that the novelty over Chow and Ghavamzadeh is not so significant. For this reason, I will keep my weak accept score, acknowledging that this a an above average submission but whose novelty is not particularly pronounced (at least in the present form of the submission).

Reviewer 2



The paper applies CVaR-based policy gradient method on robust option learning. The authors interpret the CVaR constrained objective as an unconstrained objective of a new MDP. It is interesting to see the connection and how that can simplify the understanding of the new policy gradient method. Experiments are done to understand the three questions mentioned in Section 4. However, correct me if I misunderstood, the difference between OC3 and Algorithm 1 in Chow and Ghavamzadeh[6] is just that OC3 are updating a set of \theta - {\theta_{\pi_omega}, \theta_{\beta_\omega}}_{\omega \in \Omega}, \theta_{\pi_\Omega} - yet Chow and Ghavamzadeh is updating one \theta. Other than that, the two algorithms are almost the same. Meanwhile, the MDP defined in section 3.2 is also quite similar to the augmented MDP mentioned in Section 5.1 of Chow and Ghavamzadeh [6]. Although applying existing robust MDP methods to robust option learning is important, it is unclear to me what the originality is of the proposed algorithm. Re: Author Response Thanks for clarifying my concerns. I have adapted my score accordingly. Typo: - L149: the expected loss R'' can be written as...

Reviewer 3



It seems that the assumption of P being structured as a Cartesian product is crucial for the derivation. However, it was not discussed why the assumption is valid. It seems equally possible that a distribution was chosen at the beginning and then carried through for each state. It would be helpful to explain why augmented states and rewards are necessary. Eq. 5: y should be \nv

Reviewer 4



This paper provides an interesting robust option learning formulation that's based on cost minimization and CVaR constraint, and propose an option-critic algorithm for the CVaR MDP problem. The authors also provided some derivations of robust option learning algorithms (that's based on soft robust loss function in equation 1) in the appendix. While I think these robust option-critic algorithms and derivations can potentially be solid contributions to the hierarchical RL (HRL) community, I do have the following questions/concerns: 1) While I understand the paper is proposing the CVaR option learning algorithm, how're the derivations of option-critic w.r.t. soft robust loss related to the CVaR option critic algorithm? Without looking further into the literature, it seems that the soft robust option-critic algorithm is also something that the authors are trying to propose/analyze in this paper? Are there any relationships between these analysis and the CVaR augmented MDP? 2) Without further understanding of the analysis (from the above question), it seems the contribution of this work is quite incremental. Because when the CVaR MDP is expressed as an augmented MDP, why can't one just apply Bacon's option-critic algorithm for risk-neutral MDPs to solve this problem? The CVaR MDP augmentation technique has also been proposed by Chow 2014 (which referred to some other earlier papers, such as https://link.springer.com/article/10.1007%2Fs00186-011-0367-0 for similar techniques used in the special case of finite horizon problems). So it seems the novelty here is only to extend this CVaR MDP framework to include options. 3) While the experimental results are positive in showing that CVaR option-critic outperforms most other baselines, most benchmarks are simple mujoco control task. Can the authors provide some motivations on why option-learning are needed here? I understand that for the hopper ice-block problem, using option framework probably makes sense, and I appreciate the in-depth study that is included in the Appendix. But for other tasks such as halfcheetah and walker, why can't we just solve the problem with standard CVaR PG, such as ones from Tamar 15 and Chow 14? If so, how does the performance of the vanilla CVaR PG algorithm compare with the option-critic counterpart? That being said, in general this work potentially has some good contributions to robust option learning. The following score is merely reflecting my above concerns/questions. I would consider adjusting that based on the authors' responses.

[Author Response · NeurIPS 2019]

Dear all reviewers, we highly appreciate your valuable comments and will reflect your comments in the revision.

» Revewer 1

» *"1. I think more could be done on the experimental front. ... in better RL in a number of settings."*

In all the environments except for Walker2D-disc, OC3 and SoftRobust successfully acquired options corresponding to
decomposed skills required for solving the tasks. Especially, OC3 produces robust options. In HopperIceBlock-disc,
for example, OC3 produces options corresponding to walking on slippery grounds and jumping onto a box (Figures
1 and 2). In HalfCheetah-disc, OC3 produces an option for running (highlighted in green in Figure 3) and an option
for stabilizing the cheetah-bot's body (highlighted in red in Figure 3), which is used mainly in the rare-case model
parameter setups. Acquisition of such decomposed robust skills is useful when transferring to a different domain, and
they can be also used for post-hoc human analysis and maintenance, which is an important advantage of option-based
   RL over flat-policy RL.

Figure 1: Learnt option 1 (walk)

Figure 2: Learnt option 2 (jump)

Figure 3: Learnt options (shown in green when option 1 is taking the control and in red otherwise) in Halfcheetah-cont.
Left: the case with a normative model parameter setup. Right: the case with an abnormal model parameters setup.

» *"2. The soft robust loss baseline is not that bad ... to demonstrate the benefits of the proposed framework."*

We have conducted additional experiments with the condition that the model parameter distribution and the parameter
value ranges in the test phase are more uncertain (in terms of of entropy) than those in the training phase. The results
confirmed that OC3 outperforms SoftRobust with respect to multiple performance measures (average return/loss and
CVaR) (the results will be reported in the revised version). The sim2real experiment couldn't be done in this time frame.

» Reviewer 2

» *" However, correct me if I misunderstood, ... it is unclear to me what the originality is of the proposed algorithm. "*
Let us clarify the difference between our algorithm and Algorithm 2 in [6][1]. In addition to the type of parameters you
pointed out, Algorithm 2 in [6] and our algorithm are primarily different in the form of augmented MDPs. In Algorithm
2 in [6], the augmented MDPs take the form of $\langle S', A, R', \gamma, T', \mathbb{P}'_0 \rangle$, while, in our algorithm, the augmented MDPs
take the form of $\langle S', A, R', \gamma, \mathbb{E}_p\left[T'_p\right], \mathbb{P}'_0 \rangle$. Here $T'$ is the transition function without model parameters, and the other
elements are the same as those in our paper. In sum, the types of transition function are different.

This difference comes from the difference of optimization objectives. In [6], the optimization objective is the expected
loss without a model parameter uncertainty ($\mathbb{E}_{\mathcal{R}'}\left[\mathcal{R}'\right]$), while in our optimization objective, the model parameter uncer-
tainty is introduced by taking the expectation of the model parameter distribution ($\mathbb{E}_{\mathcal{R}',p}\left[\mathcal{R}'\right] = \sum_p \mathbb{P}(p)\mathbb{E}_{\mathcal{R}'}\left[\mathcal{R}' \mid p\right]$).

You may be concerned that modifying the augmented MDPs in [6] to use $\mathbb{E}_p\left[T'_p\right]$ for dealing with the parameter
uncertainty is straightfoward. It turned out, however, that this seemingly simple modification required five pages of
proofs for justification, and any theoretical discussion for this has not been provided in previous work (e.g., in [6], [32],
and [1]). This is the reason why we described theoretical discussions (i.e., Eq. 11 $\sim$ 14 based on our option policy
gradient theorems in Appendix) for this modification, which should help many researchers in the field.

» Reviewer 3 : Thank you very much for your comments. We will modify our paper as you pointed out.

» Reviewer 4

» *"1) While I understand the paper is proposing ... is only to extend this CVaR MDP framework to include options."*
In the derivation of our CVaR option critic, the optimization of the soft robust loss $\mathbb{E}_{\mathcal{R}',p}\left[\mathcal{R}'\right]$ in augmented MDPs is
necessary (please see our answer to Reviewer 2's Q). So, as a part of the derivation, we also propose a new option critic
architecture to deal with this loss. Note that, due to the model parameter uncertainty (i.e., soft robust loss $\mathbb{E}_{\mathcal{R}',p}\left[\mathcal{R}'\right]$),
the standard option-critic framework (the Bacon's one) w. Chow's CVaR PG cannot be applied to our setting. Both of
the previous frameworks (and Tamer's framework) cannot deal with the case with model parameter uncertainty.

» *"3) While the experimental ... the vanilla CVaR PG algorithm compare with the option-critic counterpart?"*
OC3 produces some useful portable options even in Halfcheetah and Walker2D-cont (see our answer to Reviewer 1's
Q). Note that, since Chow's framework [6] (and Tamer's framework) cannot deal with model parameter uncertainty,
these cannot be applied to our setting. Finally, we will modify the notations as you suggested.

## Footnotes

[1]We assume that you mean Algorithm "2" (Actor-Critic Algorithms for CVaR Optimization), not Algorithm "1", in [6].


[Meta-Review · NeurIPS 2019]

The proposed algorithm is interesting and advances the state of the art for robust RL, however the reviewers questioned its novelty due to its high similarity with previous work. Nevertheless, the reviewers thought that the work is still valuable and technically strong.